# Osteoporotic hip fracture prediction from risk factors available in administrative claims data—A machine learning approach

**Alexander Engels**[1][*], **Katrin C. Reber**[1], **Ivonne Lindlbauer**[1], **Kilian Rapp**[2],
**Gisela Büchele**[3], **Jochen Klenk**[3], **Andreas Meid**[4], **Clemens Becker**[2], **Hans-Helmut König**[1]

1 Department of Health Economics and Health Services Research, Hamburg Center for Health Economics, University Medical-Centre Hamburg-Eppendorf, Hamburg, Germany, 2 Department of Clinical Gerontology and Rehabilitation, Robert-Bosch-Hospital, Stuttgart, Germany, 3 Department of Epidemiology and Medical Biometry, University of Ulm, Germany, 4 Department of Clinical Pharmacology and Pharmacoepidemiology, University of Heidelberg, Heidelberg, Germany

☯ These authors contributed equally to this work.
* a.engels@uke.de

**Data Availability Statement:** Data are owned by the German statutory health insurance SVLFG. To request the data please contact the institutional body of the SVLFG directly (gleichgewicht@svlfg.

## Abstract

### Objective

Hip fractures are among the most frequently occurring fragility fractures in older adults, associated with a loss of quality of life, high mortality, and high use of healthcare resources. The aim was to apply the superlearner method to predict osteoporotic hip fractures using administrative claims data and to compare its performance to established methods.

### Methods

We devided claims data of 288,086 individuals aged 65 years and older without care level into a training (80%) and a validation set (20%). Subsequently, we trained a superlearner algorithm that considered both regression and machine learning algorithms (e.g., support vector machines, RUSBoost) on a large set of clinical risk factors. Mean squared error and measures of discrimination and calibration were employed to assess prediction performance.

### Results

All algorithms used in the analysis showed similar performance with an AUC ranging from 0.66 to 0.72 in the training and 0.65 to 0.70 in the validation set. Superlearner showed good discrimination in the training set but poorer discrimination and calibration in the validation set.

### Conclusions

The superlearner achieved similar predictive performance compared to the individual algorithms included. Nevertheless, in the presence of non-linearity and complex interactions,

de). In order to fulfill the legal requirements to obtain that kind of data, researchers must conclude a contract with the SVLFG regarding data access. The licensee is permitted to use the data for the purpose of the research proposal only. Licensees are not allowed to pass the data to a third party, or to create Software or data bases with the exception of scientific publications. Moreover, the study has to be approved by the data protection officer both at the SVLFG and the research institute.

**Funding:** The study was supported by the German Federal Ministry of Education and Research (grant no. 01EC1404D).

**Competing interests:** The authors have declared that no competing interests exist.

this method might be a flexible alternative to be considered for risk prediction in large datasets.

## Introduction

Hip fractures are among the most frequently occurring fragility fractures in older adults, associated with a loss of quality of life, high mortality, and high use of healthcare resources [1]. Some of the highest hip fracture rates are seen in Europe and particularly in northern Europe–i.e., Denmark, Sweden and Norway [2, 3]. In 2010, the number of incident hip fractures in the EU has been estimated to amount to 610,000. Total costs of osteoporosis added up to about €37.4 billion. Hip fractures were determined to account for the majority (54%) of total osteoporosis costs [3].

In view of the growing population of older people, early and correct detection of those with an increased fracture risk is important to provide adequate treatment and reduce the socioeconomic impact of fractures. Approaches to fracture risk assessment such as FRAX [4], Qfracture [5, 6] or the German DVO Tool [7] are well-established to predict osteoporotic fracture risk based on various (clinical) risk factors including e.g., increasing age, female gender, low body mass index, low bone mineral density (BMD), history of fragility fractures, history of falls, smoking, alcohol intake, glucocorticoid use, other causes of secondary osteoporosis [8]. Yet, these tools rely on direct patient information to receive parameters relevant for risk prediction. In addition, these risk assessment tools often assume linear relationships between risk factors and fracture outcome. Administrative claims data have become an important source of information for payers and policymakers to support health care decision making [9]. Moreover, claims data in Germany include information, which can be difficult to obtain when questionairs or interviews are used, because–as opposed to questionairs and interviews–the data is not prone to recall (e.g., patients who do not remember their prescription dates) or recruitment biases (i.e., the full cohort of insuree's and not just those who choose to participate are assessed). Therefore, we applied a Cox proportional hazards regression in a previous analysis to assess claims data's potential to predict fracture risk [10]. However, given the number and complexity of (longitudinal) individual-level information embedded in claims data, these traditional prediction modelling techniques may be less suited to capture higher-order interaction or nonlinear effects [11]. Only recently, advanced machine learning methods–e.g., neural networks, ensembling strategies or gradient boosting–have begun to be used for clinical prediction models [12, 13]. These new techniques may have the potential to enhance risk prediction, thereby improving the chances of correctly identifying high-risk populations and offering interventions in a more efficient and targeted way.

To date, it is difficult to estimate the potential of these techniques, because they are rarely systematically compared to traditional approaches. Furthermore, the utility largely depends on where and how they are employed. Miotto, Li [14] recently showed that unsupervised feature learning based on neural networks can significantly boost the consecutive disease classification from an area under the receiver operating curve (AUC) of 0.632 (worst alternative) to 0.773. However, other studies report rather small improvements in the AUC of around 3% when comparing modern machine learning techniques to traditional approaches [15, 16]. Thus, we believe that it is important to examine whether applying more complex algorithms as opposed to traditional regression techniques offers incremental value in various contexts to learn when this additional effort amounts to meaningful improvements.

It is often not straightforward to choose a priori the "best" prediction algorithm, but super learning [17, 18], an ensemble machine learning method, can assist researchers in making this decision by combining several (pre-identified) prediction algorithms into a single algorithm.

The aim of the study was to develop and validate a prediction algorithm for osteoporotic hip fracture based on claims data employing a superlearner approach.

## Methods

### Ethics

The present study is a retrospective, observational, non-interventional study and all data were fully anonymized, therefore approval by an Ethics Committee was not required.

### Sample

In this study, we used administrative claims data from April 2008 through March 2014 on 288,086 individuals aged 65 years and older, without level of care. This datasource was also used in a previous study of ours [10]. In Germany, there were three distinct levels of care until 2016, which were clearly defined and routinely assessed by a qualified physician or nurse. The classification depended on daily time needed for care (care level 1, 2, and 3 requiring basic care such as washing, feeding, or dressing for at least 0.75, 2, and 4 hours daily time, respectively) and on whether domestic supply was necessary [19]. Individuals with care level will already have an elevated risk of falls and fracture due to a higher level of functional disability and were therefore excluded. Notably, the dataset is limited to people working in agriculture and their families, because the data provider is the German agricultural sickness fund–in german: Sozialversichung für Landwirtschaft, Forsten und Gartenbau (SVLFG). We only included individuals, who were insured by the SVLFG on April 1, 2010 (baseline) with continuous insurance coverage for the 24 months pre-period–i.e., we excluded patients who switched to the SVLFG during the pre-period or after April, 1, 2010.

### Outcome variable

The outcome variable was the first hip fracture–both osteoporotic and non-osteoporotic–occurring within 4 years after the index date–i.e., between April 1 2010 and 31 March 2014. Hospital admission and discharge diagnoses were used to identify hip fractures (International Classification of Diseases, 10th revision, German Modification, ICD-10 codes: S72.0 to S72.2).

### Predictor variables

Numerous risk factors have been identified by prior research to predict hip fracture. We used information available within administrative claims data to determine potential risk factors. Age, gender, prior fracture history, and medication use were considered as candidate predictor variables.

Age was assessed at baseline. We further assessed whether at least one prior fracture within 2 years preceding baseline was recorded (yes/no). We distinguished between prior hip fracture (ICD-10 codes: S72.0 to S72.2), prior major osteoporotic fracture (i.e., hip, vertebra, forearm or humerus fractures) [20] and prior osteoporotic fracture (vertebra, pelvic, rib, humerus, forearm, tibia and fibula, clavicle, scapula, sternum, proximal femoral and other femoral fractures) [21]. Regarding medication use, we considered exposure (yes/no) to the following risk factors: (1) drugs for which an association with fracture risk has been well established, e.g., glucocorticoids, aromatase inhibitors, antidepressants, proton pump inhibitors [4, 6, 7], (2) drugs commonly prescribed for conditions that have been associated with increased fracture risk, e.g.,

antidiabetics, and (3) drugs prescribed for prevention and treatment of osteoporosis such as bisphosphonates, calcium, vitamin D and their combinations. Exposure was defined as at least two prescriptions recorded in the seven months before baseline [6, 22, 23]. The validity and reliability of outpatient diagnoses recorded in claims data are limited [24, 25]. Thus, we decided to exploit the information on prescribed medications as a surrogate for diseases/disorders associated with increased fracture risk. Henceforth, we refer to these predictors as "drug-related risk factors".

## Analysis methods

We applied the superlearner (SL) approach [17, 18, 26] to predict the occurrence of hip fracture within 4 years of baseline. Superlearning is an ensemble machine learning method for choosing via cross-validation the optimal weighted combinations of the predictions made by a set of candidate algorithms. It does not require an a priori selection of algorithms, but is technically capable of selecting the best set of algorithms from multiple options and integrating the results from the relevant ones. Candidate algorithms can be both parametric and non-parametric and each algorithm is k-fold cross-validated on a dataset to avoid overfitting. We applied 10-fold cross-validation, which divided the dataset into $k = 10$ mutually exclusive and exhaustive sets of almost equal size. For each $k$ fold, one of the k sets serves as validation set, the others act as training sets. Each algorithm is fitted on the training set to construct the estimators whose performance (so-called risk or squared error) is then assessed in the validation set. Since overly flexible algorithms tend to exploit random variation in the training data to increase accuracy, the performance has to be assessed for the validation dataset. The process is repeated until each set has served both as training and validation sample and predicted values are obtained for all observations. Simple regression techniques may then be used to determine the utility (i.e., the beta-coefficients) of the predicted values of the algorithms for predicting the outcome. Non-significant predictions and their respective algorithms are excluded. A new estimator (so-called SL-estimator) is then generated as a weighted combination of the relevant predictions from the candidate algorithms that yields the smallest squared prediction error. Ideally, the algorithms should be heterogeneous in their statistical properties (i.e., some ought to be parsimonious while others ought to be flexible), in order to allow for different levels of complexity in the data.

Furthermore, we compare our model with extreme gradient boosting (XGBoost). XGBoost is a comprehensive and versatile library, which offers a powerful framework for implementing Gradient Boosted Trees (GBTs). These build an ensemble of multiple weak trees (e.g., trees with few decision rules) in sequence, thereby allowing each tree to learn and improve upon the previous trees. It is a state of the art machine learning approach that outperformed traditional techniques in various settings [27, 28]. Therefore, it is currently the best option for a gold standard comparison. Details on the parameter values of the final model and how they were obtained can be found in the supplement.

## Candidate learning algorithms for the superlearner

We considered the following candidate learning algorithms: Logistic regression using forward and backward variable selection (main effects only) [29], random forests [30], support vector machines (SVM) [31] and RUS (random undersampling)—Boost with SVM as learner [32, 33]. Additionally, we considered the stepwise logistic regressions with an alternative model specification that included two-way interaction between age and all other predictors as well as sex and all other predictors. Random forests (RF) combine predictions from all regression or classification trees that have been fitted to a data set. The growth of each tree is based on a

random process, which uses a randomly drawn subsample and a random subset of the available features for each splitting decision. Thus, the method requires a large number of individual trees to detect the most important variables and make accurate predictions.

SVM aim to classify cases by constructing a hyperplane that achieves the best partitioning of the data by maximizing the margin between the closest points of two classes. Whenever a linear separator cannot be found, the observations are mapped to a higher-dimensional space using a (non-)linear kernel function to enable linear separation [34].

RUSBoost, a hybrid approach designed for imbalanced data problems, combines random undersampling and boosting. The latter generates a strong classifier from a number of so-called weak learning algorithms. These weak learners ought to achieve accuracy just above random chance. We chose the AdaBoost.M2-algorithm [35] using a support vector machine with a linear kernel as weak learner. AdaBoost applies a weak learner repeatedly to predict the most fitting class. A set of plausibility values for the possible classes is assigned to each case. The weak learners are evaluated using a loss-function that penalizes different types of misclassification. With each iteration, the loss-function values are updated allowing the algorithm to focus on classes which are particularly difficult to distinguish from the correct class. By addressing these difficult cases, AdaBoost.M2 can outperform other methods in imbalanced datasets, where the correct classification of the minority class is often most challenging. An overview of the algorithms is provided in the electronic supplement (S1 File of S1 Table).

## Random undersampling

The dataset was divided into a training (80%) and a validation dataset (20%). For the training set, random undersampling methods were applied to address that most algorithms try to minimize the overall error rate. In our context, predicting exclusively non-fractures would already result in an extremely low error rate, although the predictions would practically be useless. Thus, although random undersampling is associated with a loss of information [36], it may improve the classifiers' performance with regard to the AUC [37] by reducing the overwhelming influence of the majority class in imbalanced datasets. In addition, it has been shown that methods that rely on random undersampling in an imbalanced setting are often more simple, faster, and comparable (if not better) in their performance compared to other sampling methods [33]. In a first step, a one-sided selection method was used [38] that eliminates cases from the majority class (*no fracture*) while keeping all cases from the minority class (*fracture*). In a second step, random undersampling was performed until the ratio of minority to majority class was 3:7. Given that the base rate of hip fractures in the original dataset was only about 3%, complete balance, (i.e., a ratio of 1:1) was deemed too expensive, because the number of cases lost due to undersampling substantially increases the more balanced the ratio becomes. The final training set resulted in 20,456 individuals. Each candidate algorithm was implemented using the (undersampled) dataset. The predictors considered in the analyses are listed in Table 1.

We applied the SL approach (as described above) to select the best combination of these algorithms specifying 10-fold cross-validation. Superlearner is implemented in the R package Super Learner [39]. All analyses were carried out using R version 3.4.1 (R Foundation for Statistical Computing, Vienna, Austria).

## Evaluating algorithm performance

All candidate algorithms that enter the superlearner were fitted on the entire training dataset and their performance further assessed using the validation dataset. To evaluate classification performance the mean squared errors (MSE) and AUC values were calculated for both the

**Table 1. Characteristics of study population (n = 288,086).**

| Characteristic | | No. | | % |
|---|---|---|---|---|
| Female gender | | 140,709 | | 48.8% |
| Age (at baseline), years | Mean (SD) | 75.67 | (6.20) | |
| Hip fracture within the 4 year follow-up | | 7,644 | | 2.7% |
| Patients without a hip fracture within the 4 year follow-up | | 280,442 | | 97.3% |
| Patients without a hip fracture within the 4 year follow-up who were not lost to follow-up | | 231,578 | | 80.4% |
| Prior osteoporotic fracture (2 years) | | | | |
| all | | 7,032 | | 2.4% |
| minor | | 2,580 | | 0.9% |
| major | | 4,864 | | 1.7% |
| hip | | 1,854 | | 0.6% |
| Medication (within the seven months before baseline): | | | | |
| Antiparkinson agents | | 7,050 | | 2.4% |
| Anticonvulsants/Antiepileptics | | 8,313 | | 2.9% |
| Aromatase inhibitors | | 1,295 | | 0.4% |
| Antidiabetic agents | | 36,782 | | 12.8% |
| Proton pump inhibitors | | 55,770 | | 19.4% |
| Antidementives | | 2,509 | | 0.9% |
| Drugs for obstructive airway diseases | | 29,616 | | 10.3% |
| Bisphosphonates | | 9,451 | | 3.3% |
| Bisphosphonate combinations | | 1,831 | | 0.6% |
| Raloxifene | | 304 | | 0.1% |
| Antidepressants, psycholeptics, and their combinations | | 42,849 | | 14.9% |
| Gestagens, estrogens, and their combinations | | 10,030 | | 3.5% |
| Glucocorticoids (systemic), and combinations with antiphlogistics/antirheumatics | | 20,648 | | 7.2% |
| Anti-inflammatory and antirheumatic agents | | 2,299 | | 0.8% |
| Calcium, vitamin D and analogues, and combinations | | 9,798 | | 3.4% |
| Thyreostatic agents | | 3,632 | | 1.3% |
| GnRH analogues, antiandrogens | | 3,775 | | 1.3% |
| Ophthalmic agents | | 33,351 | | 11.6% |
| Anticholinergic agents | | 7,786 | | 2.7% |
| Tamsulosin | | 20,787 | | 7.2% |
| Lost to follow-up: | | | | |
| Total (death within four years, other reasons) | | 51,476 | | 17.9% |
| Death within the first year | | 7,721 | | 2.7% |
| Death within twoyears | | 17,492 | | 6.1% |
| Death within three years | | 28,957 | | 10.1% |
| Death within four years | | 40,527 | | 14.1% |

GnRH, Gonadotropin-releasing hormone.

individual algorithms and the superlearner. AUC is used to evaluate overall prediction accuracy. Model calibration was assessed with the Hosmer-Lemeshow statistic [29]. Moreover, we used calibration plots, which plot actual fracture percentages against those predicted by the algorithms for various risk quantile to assess (lack of) fit. Due to random undersampling, the base rate in the training set differed from the base rate in the validation set. Therefore, the predicted probabilities were adjusted before using them for model calibration in order to avoid overestimation of the proportion of fracture events [40].

## Results

### Descriptive analysis

A summary of variables used in the analysis is presented in Table 1. Of the 288,086 individuals that were originally included in the sample, 20,456 were used during training and 57,618 in the validation dataset. Mean age of the original sample was 75.7 years (SD: 6.2), 48.8% were female. About 3% sustained a hip fracture during follow-up. 2.4% of the sample had any prior clinically diagnosed osteoporotic fracture and 0.6% had a prior diagnosed hip fracture. Drug-related risk factors affecting the largest percentage of the sample were proton pump inhibitors (19.4%), followed by antidepressants, psycholeptics, and their combinations (14.9%), antidiabetic agents (12.8%), ophthalmic agents (11.6%), and drugs for obstructive airway diseases (10.3%).

### Model performance

The performance of the superlearner was similar to other individual algorithms used in the analysis. With regard to the Brier score, the superlearner algorithm for predicting hip fracture improved upon random forests by 6%. The superlearner performed very similarly to RUS-Boost with only marginal improvement in Brier score. Compared to logistic regression the superlearner performed slightly worse with respect to brier score. All algorithms achieved moderate discriminatory performance, with AUC values ranging from 0.650 for SVM to 0.704 for logistic regression in the validation set and from 0.660 for SVM to 0.721 for superlearner in the training set. The superlearner (AUC 0.698, 95% CI 0.684–0.711) was slightly outperformed by logistic regression (AUC 0.704, 95% CI 0.691–0.718) and XGBoost (AUC 0.703, 95% CI 0.689–0.716) in the validation set. In the training set, the superlearner and XGBoost performed better than the candidate algorithms regarding their discriminatory ability, with an AUC of 0.722 (95% CI 0.714–0.729) for the superlearner and an AUC of 0.725 (95% CI 0.718–0.733) for XGBoost. Results are shown in Table 2. Furthermore, we provide information on the number of false negatives and false positives as well as the corresponding rates for the superlearner and our benchmark model XGBoost in the S1 File of S1 and S2 Figs.

Actual and predicted fracture percentages are shown in Fig 1. The calibration plot indicates that the superlearner (right panel) underestimated actual fracture probability to some extent. Thus, the fit for the superlearner was only moderate (Chi-Square = 154.75, p<0.001).

**Table 2. Brier score and AUC for each algorithm.**

| Algorithm | Brier score | Validation | | Training | |
|---|---|---|---|---|---|
| | | AUC | (95% CI) | AUC | (95% CI) |
| Logistic regression with forward selection | 0.0251 | 0.704 | (0.691–0.718) | 0.713 | (0.705–0.720) |
| Logistic regression with forward selection and interactions | 0.0265 | 0.698 | (0.685–0.712) | 0.712 | (0.705–0.720) |
| Logistic regression with backward selection | 0.0261 | 0.704 | (0.690–0.717) | 0.713 | (0.705–0.720) |
| Logistic regression with backward selection and interactions | 0.0267 | 0.695 | (0.681–0.708) | 0.714 | (0.706–0.721) |
| Random forest | 0.0268 | 0.685 | (0.671–0.699) | 0.686 | (0.678–0.694) |
| Support vector machines | 0.0252 [a] | 0.650 | (0.635–0.666) | 0.660 | (0.651–0.668) |
| RUSBoost | 0.0254 | 0.702 | (0.688–0.715) | 0.711 | (0.703–0.718) |
| Superlearner | 0.0259 | 0.698 | (0.684–0.711) | 0.722 | (0.714–0.729) |
| XGBoost | 0.0251 | 0.703 | (0.689–0.716) | 0.725 | (0.718–0.733) |

CI, confidence interval.

[a] Brier score was calculated by transforming the support vector machine output to probabilities using a sigmoid link function.

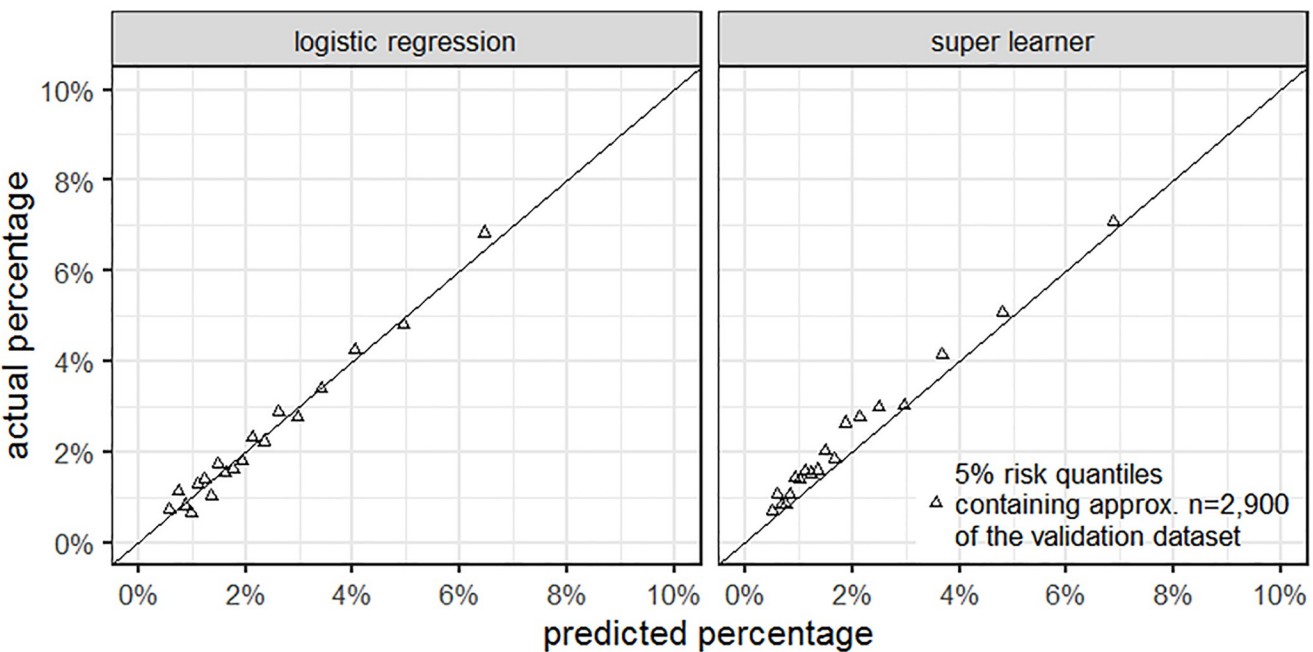

**Fig 1. Calibration plots for logistic regression (left panel) and superlearner (right panel).** The plots show the calibration for the validation dataset. We grouped the n = 57,618 individuals in the validation dataset according to their respective 5% risk quantile as predicted by either the logistic regression or the superlearner. For each quantile group, we plotted the predicted proportion of S72.0-S72.2 fractures against the actual proportion of S72 fractures.

However, for logistic regression, which performed best as regards Brier score and AUC in the validation set, calibration was good (Chi-Square = 15.14, p = 0.06).

## Components of risk

The output of machine learning techniques is not as readily interpretable as the significance tests for the coefficients of parametric techniques. The performance regarding prediction accuracy can easily be assessed, while the importance of specific variables cannot be determined without additional analysis. Thus, we used the logistic model (with forward selection) to determine which variables were important for fracture prediction. In particular, female gender, older age, and prior fracture history were associated with a higher probability of sustaining hip fractures. Also, several medical conditions (operationalized by medication use) such as Parkinson's disease, dementia, or diabetes as well as the use of glucocorticoids were significantly associated with higher fracture probability (Table 3).

## Discussion

In this study, we developed a prediction algorithm to assess the risk of osteoporotic hip fractures using claims data. The focus was on machine learning techniques including both traditional and newer approaches. In particular, we applied the ensembling superlearner algorithm that can be employed in large administrative claims databases for fracture prediction.

We found the performance of the superlearner algorithm to be similar to the individual algorithms used in the analysis. Although the superlearner did perform no worse than the candidate algorithm in the training set, in the validation set logistic regression, XGBoost and RUS-Boost had similar or higher AUCs. Furthermore, when comparing the results of this study

**Table 3. Coefficients of multivariable logistic regression to predict hip fracture.**

| Predictor | β value | SE | p |
|---|---|---|---|
| Intercept | -9.216*** | 0.204 | 0,000 |
| Age (effect for each additional year) | 0.101*** | 0.003 | 0,000 |
| Female gender | 0.628*** | 0.034 | 0,000 |
| Prior osteoporotic fracture (2 years) | 0.402*** | 0.099 | 0,000 |
| Prior osteoporotic hip fracture (2 years) | 0.635*** | 0.175 | 0,000 |
| Antidiabetic agents | 0.276*** | 0.047 | 0,000 |
| Antiparkinson agents | 0.385*** | 0.092 | 0,000 |
| Antidementives | 0.482*** | 0.138 | 0,000 |
| Anticonvulsants/Antiepileptics | 0.270** | 0.087 | 0,002 |
| Proton pump inhibitors | 0.109** | 0.040 | 0,007 |
| Antidepressants, psycholeptics, and their combinations | 0.096* | 0.044 | 0,028 |
| Anticholinergic agents | 0.208* | 0.091 | 0,022 |
| Antiinflammatory and antirheumatic agents | 0.338* | 0.167 | 0,042 |
| Aromatase inhibitors | 0.319 | 0.218 | 0,144 |
| Thyreostatic agents | -0.182 | 0.130 | 0,159 |
| Glucocorticoids, and combinations with antiphlogistics/antirheumatics | 0.198*** | 0.061 | 0,001 |
| GnRH analogues, antiandrogens | 0.338** | 0.125 | 0,007 |
| Gestagens, estrogens, and combinations | -0.244** | 0.091 | 0,008 |
| Bisphosphonates | 0.212** | 0.082 | 0,009 |
| Bisphosphonate combinations | 0.437* | 0.184 | 0,018 |

Results are based on the undersampled training dataset. Therefore, a meaningful interpretation can only be given for the effect direction and not the magnitude.

*** $p \leq 0.001$

** $p \leq 0.01$

* $p \leq 0.05$. SE—standard error; GnRH—Gonadotropin-releasing hormone.

with the results of a previous publication, in which we used a Cox proportional hazard model for the prediction of fracture risk [10], we only find negligible differences in the predictive performance. Consequently, the machine learning approach offers no benefits in this context when compared to traditional approaches. Regarding our data source, we found that predictions based on german claims data are considerably worse than predictive models that take advantage of clinical information such as bone mineral densities and biochemical glucose measurements. In Denmark, a recent study reported an AUC of up to 0.92 for hip fracture predictions, which highlights the benefit of using a National Patient Registry that contains clinical and laboratory information for risk prediction [41].

Notably, in several datasets the superlearner increased performance when compared to its candidate algorithms [18]. Nevertheless, we found multiple studies, who correspondingly found no major differences between the supe learner and more traditional methods or even somewhat worse performance [15, 42].

In this study, the slightly poorer calibration of the superlearner may be partly due to the fact that the algorithm did not estimate probabilities. Thus, the adjustment method proposed by Saerens et al. [40] may have been imperfectly suited for incorporating the bias due to random undersampling.

As it is recommended to decide upon input variables, candidate algorithms, and their specification before running the analysis [11], we considered risk factors commonly included in fracture risk assessment tools that were also available in claims data. To help to achieve a better

understanding of important risk factors, we reported the output of a logistic regression analysis and report a variable importance plot of XGBoost in the supplement. However, caution is needed in interpreting the results of an exploratory logistic regression with a forward selection method, because it exploits random variation. Nonetheless, it appeared that next to some drug-related risk factors, risk factors such as gender, prior fracture, and age noticeably contributed to risk prediction. Thus, the model performance was mainly dependent on few predictors that are known to be important. This is in line with other studies finding so simple models consisting of fewer predictors e.g., age, gender, and prior fracture perform as well as more complex models in predicting fracture [43, 44].

The fact that drug-related risk factors had limited additional explanatory power in this study may partly explain moderate performance of the superlearner. Our choice to employ a superlearner approach in this study was driven by the effectiveness of the technique in other studies [12, 18], the low base rate of hip fractures in our data and the relatively large number of potential predictors we initially considered. It is still not fully understood whether and when complex ensemble machine learning techniques offer real value over traditional methods in clinical prediction tasks [45]. We agree with previous findings on this question that a strong non-linear relationship between the predictors and the response appears to be an important prerequisite [46] and that the signal-to-noise ratio ought to be high [47].

## Strengths and limitations

Major strengths of the study are the large number of observations and the rich set of potential risk factors derived from administrative claims data. This allowed us to apply powerful machine learning methods for fracture prediction, which have the potential to take advantage of complex interactions and unknown non-linear effects. Moreover, we performed a systematic fracture outcome ascertainment based on ICD-10 hospital diagnoses. Hence, outcome misclassification, which often is an issue in ambulatory settings, should be low as nearly all people with hip fractures are admitted to hospital.

Since we were faced with highly imbalanced data as regards fracture events, we employed random undersampling to mitigate the issue that some algorithms perform poorly in imbalanced datasets. However, we are well aware that there are alternative measures for dealing with this issue. For instance, we could have applied oversampling, which would entail replicating the minority samples until fracture and non-fracture events are well-balanced. In our study, this would require n = 120,189 minority cases–i.e., we would need to replicate each minority case almost 16 times–to reach the desired ratio of 3:7. Drawing each minority case that frequently substantially increases the risk of overfitting. Moreover, given our sample size, oversampling would drastically increase the computational runtime and the probability for running out of memory. Consequently, we decided on random undersampling, although randomly drawing from the majority sample sometimes discards informative cases [48, 49] and it can distort significance tests and the magnitude of the parameter estimates–i.e., the obtained estimates and p-values ought to be interpreted with caution. In the future, it might be a promising alternative to optimize other metrics than the global classification accuracy, because non-decomposable functions such as the F-Score or precision are more natural choices for imbalanced data that do not require random over- or undersampling [50]. Unfortunately, current implementations are limited to using these functions as evaluation criteria, which does not enable parameter optimization.

We recognise that the ascertainment of (co-)morbidities proves a challenge in (administrative) ambulatory claims data. The lack of reliability of diagnosis coding in the ambulatory setting has been repeatedly shown [24, 25]. Therefore, we solely considered drug exposure as surrogate variable, but no ambulatory ICD-10 diagnoses to identify and define conditions that

have been associated with increased fracture risk such as Parkinson's disease [51]. Prescribed medications whose ATC Classification starts with N04 are almost exclusively used to manage Parkinson's disease and are therefore a reliable indication for that particular disorder. This prescription based measure was chosen, because using the most reliable source of information (i.e., hospital diagnoses) would have introduced considerable bias, because in this case only individuals with a hospital stay would have been recorded. In addition, we included medications that have multiple indications but were consistently associated with fracture risk in previous studies such as psychotropic drugs [52].

Only risk factors available in administrative claims data could be considered. Risk factors drawn from self-reported data, registries/EHR such as smoking, alcohol use or BMI were not accounted for. Such information is either unavailable or cannot be retrieved from administrative claims data without bias. Including such additional risk factors might have contributed to better performance of the algorithms.

In this study, we assessed a relatively long follow-up period, because we needed to identify a sufficient number of fracture events (the incidence rate of hip fracture was only 0.6% within the first year). As a result, we were able to apply data hungry machine learning techniques [53]. Nonetheless, this approach had some disadvantages. First, almost 18% of our sample were not observed until the end of the 4 year follow-up period. In spite of that we decided to not exclude patients that switched to another statutory health insurance or passed away during the follow-up period, because this information would not be available in a prediction model that is implemented to identify insurees who will be at risk in the future. Second, we are aware that recently assessed prescription based risk factors have a higher predictive value [54]. Thus, the long follow-up period could have negatively influenced the model performance.

Finally, our data were derived from only one health insurer and may not be representative for the whole German population. Compared with persons insured by other health insurance providers, persons insured at this particular health insurer represent those living in more rural regions and working or having worked in the agricultural (incl. forestry and horticulture) sector. This may have influenced the number of hip fractures because lower rates for both hip fractures and fractures at other sites typically associated with osteoporosis have been reported in rural populations [55, 56].

## Conclusion

In general, the performance of the superlearner was similar to the included individual algorithms used in the analysis. It showed good discrimination in the training data set, but poorer discrimination and calibration in the validation set compared to the candidate algorithms. The lack of substantive difference between these methods does not speak against the superlearner per se. In other cases, the superlearner has proven to perform at least as well or even better than its candidate methods. In our case, however, any of the methods we included, and in particular simpler ones, may be used for these data to predict fracture risk.

## Supporting information

**S1 File.**
(DOCX)

## Acknowledgments

The authors thank the SVLFG and especially Daniel Stöger and Andrea Grunz from the SVLFG for granting access to the data and data support. Moreover, we thank the participants

at the 10th annual meeting of the German Health Economics Association for advice and comments.

## Author Contributions

**Conceptualization:** Katrin C. Reber, Ivonne Lindlbauer, Kilian Rapp, Hans-Helmut König.

**Data curation:** Katrin C. Reber, Ivonne Lindlbauer.

**Formal analysis:** Alexander Engels.

**Funding acquisition:** Kilian Rapp, Hans-Helmut König.

**Methodology:** Alexander Engels, Katrin C. Reber, Gisela Büchele, Jochen Klenk.

**Project administration:** Kilian Rapp, Clemens Becker, Hans-Helmut König.

**Supervision:** Hans-Helmut König.

**Validation:** Alexander Engels, Katrin C. Reber.

**Writing – original draft:** Alexander Engels, Katrin C. Reber.

**Writing – review & editing:** Ivonne Lindlbauer, Kilian Rapp, Gisela Büchele, Jochen Klenk, Andreas Meid, Clemens Becker, Hans-Helmut König.

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
