## [Decision Letter · Decision Letter 0]

27 Jan 2020

PONE-D-19-26721

Osteoporotic hip fracture prediction from risk factors available in administrative claims data – a machine learning approach

PLOS ONE

Dear Mr. Engels,

Thank you for submitting your manuscript to PLOS ONE. After careful consideration, we feel that it has merit but does not fully meet PLOS ONE’s publication criteria as it currently stands. Therefore, we invite you to submit a revised version of the manuscript that addresses the points raised during the review process.

The manuscript has been reviewed by two external experts, who both ask for some modifications to be made, please see their detailed commends below.

We would appreciate receiving your revised manuscript by Mar 12 2020 11:59PM. To enhance the reproducibility of your results, we recommend that if applicable you deposit your laboratory protocols in protocols.io, where a protocol can be assigned its own identifier (DOI) such that it can be cited independently in the future. For instructions see: http://journals.plos.org/plosone/s/submission-guidelines#loc-laboratory-protocols

We look forward to receiving your revised manuscript.

Kind regards,

Lars Kaderali

Academic Editor

PLOS ONE

2. We noticed you have some minor occurrence of overlapping text with the following previous publications, which needs to be addressed: Reber, Katrin C., et al. "Development of a risk assessment tool for osteoporotic fracture prevention: A claims data approach." Bone 110 (2018): 170-176. In your revision ensure you cite all your sources (including your own works), and quote or rephrase any duplicated text outside the methods section. Further consideration is dependent on these concerns being addressed

Reviewers' comments:

Reviewer's Responses to Questions

**Comments to the Author**

1. Is the manuscript technically sound, and do the data support the conclusions?

Reviewer #1: Yes

Reviewer #2: Yes

2. Has the statistical analysis been performed appropriately and rigorously? 

Reviewer #1: Yes

Reviewer #2: Yes

3. Have the authors made all data underlying the findings in their manuscript fully available?

Reviewer #1: Yes

Reviewer #2: Yes

4. Is the manuscript presented in an intelligible fashion and written in standard English?

Reviewer #1: Yes

Reviewer #2: Yes

5. Review Comments to the Author

Reviewer #1: The authors are presenting results on the prediction of osteoporotic hip fractures using various machine leaning techniques (logistic regression with forward selection, random forest, support vector machine, adaptive boosting with random undersampling) and compared the individual results to a meta learner/ super learner making use of all base learner. The data base consists of administrative claims from a German health insurance including more than 280000 individuals without the need for basic care. Hip fracture was recorded in a 4-year follow up and possible predictors entered the models from the baseline (e.g. age, gender, prior fractures, drugs). The authors achieved an AUROC of about 0.7.

The authors presented their results in an intelligible fashion and the paper is well written. It contains necessary data description and supporting information on the methods (with packages used in R) to allow reproducibility. The methods are well summarized.

Besides the good overall impression, I have some questions and remarks which should be addressed and which would further improve the paper:

1. Data

a) It could be of importance to add the working status at baseline as another predictor, since you are including individuals aged 65 years and older. Furthermore, I advise to include the number of drugs prescribed per person as a continuous variable as well as the the number of drugs related to osteoporosis and number of drugs not related to osteoporosis.

b) Are there other exclusion criteria than described? I am thinking of individuals passing away during the 4-year follow-up, individuals who switched to another health insurance during the follow-up (you only wrote, that you excluded cases with incomplete 24-month pre-baseline).

c) Is your response variable the all-cause hip fracture or do you distinguish between osteoporotic and non-osteoporotic hip fractures?

d) It would be nice to see the prediction performance on time scales of the outcome variable: hip-fracture occurring e.g. within 6 months, 12 months, 24 months and 48 months, since the drug prescriptions could have a higher predictive value within the first 6 months than within 48 months.

2. Methods

a) You should think of including other boosting methods such as XGBoost and Gradient boosting to allow a better comparison of current (state-of-the-art) ML techniques.

b) Under-sampling should be compared with other methods such as Up-sampling (e.g. with SMOTE). Do you also repeated the random sampling to create multiple training sets?

c) Have you tried to drop the sampling strategy and tried to optimize the F1-Score instead? The F1-Score is independent of balancing. Please do add the numbers of false positive and false negative in the validation set.

d) Think of including interactions (with age or gender) in your regression models.

Line-by-line comments on the manuscript:

65: Germany does not have one of the highest hip-fracture rates – its rather midfield, especially in the 10y probability of major fractures. Better refer to the high rates in Northern Europe (Denmark, Sweden, Norway).

66: In the original paper it is 610,000.

68: Punctuation error before reference number

82: Cox proportional hazards regression

86-87: What do you mean with ‘advanced’ ML methods – please name those

87: ‘may have the potential’ – How large is the expected effect on the AUC? Please refer to other papers on disease predictions where those new methods outperformed classical approaches.

97: The present study is

109: Is this the LKK (Landwirtschaftliche Krankenkasse)? I havn’t found ‘German agricultural sickness fund’.

143: How many folds have you used? Define k.

167: SVM already abbreviated

170: Which kernel have you used?

171: RUSBoost is an abbreviation, please add also the full name.

196: Why do you use 3:7 sampling? I there any recommendation giving in the literature?

219: I just see 20,456 individuals used for training and 20% of 288,000 for testing. To the analysis did not include 288,086 individuals.

221: Please write the number in numeric form: 2.4%

223: 0.6% of all samples

234: Please add columns with the number of individuals with and without 4yr-hip-fracture to allow univariable comparisons (and probably important for inclusion in meta-analysis).

249: The MSE for SVM can be calculated in R using caret, see https://www.quora.com/How-can-I-calculate-the-mean-square-error-MSE-for-SVM-in-R

257: Suggested section title: Predictors

267: Table header suggestion: Coefficients of multivariable logistic regression to predict hip fracture

267: Please add the p-values.

267: Please add the unit of ‘Age’ to make clear that this is the beta-coefficient for 1 year difference.

282: Full stop after algorithms and add citation

293: I disagree, this is not computationally expensive – it can be directly extracted from the model

302: Please compare your results with described performances in the literature, … AUC in model with bone density information is 0.82 …

306: Please add citation

313: If your intention was to find complex interaction, why do you just present results from the logistic regression without any interaction?

319: I don’t see any memory problems arising from a data frame with 20000 individuals and probably less than 100 features.

359: software

Reviewer #2: Review for manuscript titled with “Osteoporotic hip fracture prediction from risk factors available in administrative claims data – a machine learning approach”

This retrospective study applied several algorithms , including super learner method to predict osteoporotic hip fracture using administrative claims data and found out the the super learner achieved similar predictive performance compared to other algorithms

Introduction:

1. Well-written

2. The sentence started from the line 80 “ Moreover, claims data in Germany ……within the observational period” was not clear, please clarify

Methods:

1. In the paragraph of “sample”, the baseline for the study was not well defined. For an example, in line 108, the baseline is April, 1 2010 and pre baseline is 24 months. Were patients allowed to enter the cohort after April, 1, 2010?

2. In the paragraph of “ Outcome variable”, “hip fracture occurring within 4 years of baseline” was the outcomes of interest? Please clarify baseline and prebaseline.

3. It was not clear whether the patients were allowed to have multiple fractures or only the first one.

4. Please explain why some variables were evaluated using 2 years pre baseline, but some factors were evaluated in the seven month before baseline.

5. This paper used prescribed medications as a surrogate for diseases associated with fracture risk, however, many medications have multiple indications. Has these surrogates been validated yet?

6. Suggest to also use logistic regression backward as it is has been reported more conservative

7. The analysis section is clear and well written

Results:

1. Suggest the authors to add the time period for the medications in table 1

2. Please also provide the characteristics for patients risk during the follow up time.

6. PLOS authors have the option to publish the peer review history of their article (what does this mean?). If published, this will include your full peer review and any attached files.

Reviewer #1: Yes: Marcus Vollmer

Reviewer #2: No

---

## [Author Response · Author response to Decision Letter 0]

16 Mar 2020

We refer to the uploaded word document for a better formatted version of these responses.

Comments of Reviewer 1)

a) It could be of importance to add the working status at baseline as another predictor, since you are including individuals aged 65 years and older. 

Furthermore, I advise to include the number of drugs prescribed per person as a continuous variable as well as the the number of drugs related to osteoporosis and number of drugs not related to osteoporosis.

We agree that the employment status could be of importance. Unfortunately, this information could not be provided by our contract partner (i.e. the statutory health insurance SVLFG).

Regarding the suggestions to add the number of drugs prescribed, we believe that this would not be in line with our methodological decision to focus on risk factors that were previously shown or suggested to be associated with osteoporotic fractures in the literature or in validated fracture tools such as QFracture and FRAX. 

The list of included drugs was discussed and further refined with a clinical expert (Kilian Rapp) and a trained pharmacologist (Sarah Mächler) to ensure face validity with regard to clinical relevance and appropriate Anatomical Therapeutic Chemical (ATC) classification. While including a larger set of predictors as well as aggregates of existing ones may improve model performance, we argue that limiting the set of predictors to those with a strong theoretical foundation improves transparency and reduces the risk of overfitting (1). 

b) Are there other exclusion criteria than described? I am thinking of individuals passing away during the 4-year follow-up, individuals who switched to another health insurance during the follow-up (you only wrote, that you excluded cases with incomplete 24-month pre-baseline).

The prediction model was designed as a way for insurance companies to predict at risk groups. Successfully identifying these groups would enable efficient prevention. Consequently, it is important to not select individuals based on how they will develop in the follow-up period, because the future development (e.g., whether patients will pass away or switch to another health plan) would not be known when the prediction tool is implemented.

However, we agree that it is important to report the number of patients that were lost-to-follow-up due to death or other reasons, because the amount of censoring affects model performance. Therefore, we added this information in table 1. We also included the number of patients that die within the first, second, third and fourth year. 

In addition, we discuss the issue of decreased model performance in a new section referenced in our answer 

to bullet point (d).

c) Is your response variable the all-cause hip fracture or do you distinguish between osteoporotic and non-osteoporotic hip fractures?

All S72 femur fracture were assessed. We clarified this under the section: Outcome Variable

The outcome variable was the first hip fracture – both osteoporotic and non-osteoporotic – occurring within 4 years after the index date – i.e. between April 1 2010 and 31 March 2014.

d) It would be nice to see the prediction performance on time scales of the outcome variable: hip-fracture occurring e.g. within 6 months, 12 months, 24 months and 48 months, since the drug prescriptions could have a higher predictive value within the first 6 months than within 48 months

We used a relatively long follow-up period, because hip fracture occur rather rarely. The incidence rate was very low. For instance, within the first 6 month we observed an incidence rate of 0.27% for the sample. In order to have a sufficient number of minority cases (i.e. fracture events), it was necessary to assess a relatively long follow-up period. However, we agree that the predictive performance might be better for shorter follow-up periods. We discuss this aspect in our revised discussion.

In this study, we assessed a relatively long follow-up period, because we needed to identify a sufficient number of fracture events (the incidence rate of hip fracture was only 0.6% within the first year). As a result, we were able to apply data hungry machine learning techniques (53). Nonetheless, this approach had some disadvantages. First, almost 18% of our sample were not observed until the end of the 4 year follow-up period. In spite of that we decided to not exclude patients that switched to another statutory health insurance or passed away during the follow-up period, because this information would not be available in a prediction model that is implemented to identify insurees who will be at risk in the future. Second, we are aware that recently assessed prescription based risk factors have a higher predictive value (54). Thus, the long follow-up period could have negatively influenced the model performance. 

e) You should think of including other boosting methods such as XGBoost and Gradient boosting to allow a better comparison of current (state-of-the-art) ML techniques.

Thank you for this suggestion. We decided to implement XGBoost as our benchmark model and added a paragraph on the method:

Furthermore, we compare our model with extreme gradient boosting (XGBoost). XGBoost is a comprehensive and versatile library, which offers a powerful framework for implementing Gradient Boosted Trees (GBTs). These build an ensemble of multiple weak trees (e.g. trees with few decision rules) in sequence, thereby allowing each tree to learn and improve upon the previous trees. It is a state of the art machine learning approach that outperformed traditional techniques in various settings (2, 3). Therefore, it is currently the best option for a Gold-Standard comparison. Details on the parameter values of the final model and how they were obtained can be found in the supplement. 

f) Under-sampling should be compared with other methods such as Up-sampling (e.g. with SMOTE). Do you also repeated the random sampling to create multiple training sets?

The main tradeoff between random under- or oversampling is that oversampling may lead to overfitting as it makes exact copies of the minority samples, while undersampling may discard potential useful majority samples (4). Moreover, oversampling substantially increases computational cost (5).

Given the large sample size, we assumed that overfitting would be a larger issue than a non-representative majority sample. Moreover, we were constrained by our computational resources, which were not sufficient to compare various sampling strategies and parameter settings. Comparing multiple algorithms, using an ensembling strategy and being able to use 10-fold cross validation on our training sample was a top priority. As these methods would be computationally very expensive in an up-sampling context (given our large sample size) or in a setting where multiple training sets would have been created, we decided against it. 

However, we added this limitation to our discussion:

Since we were faced with highly imbalanced data as regards fracture events, we employed random undersampling to mitigate the issue that some algorithms perform poorly in imbalanced datasets. However, we are well aware that there are alternative measures for dealing with this issue. For instance, we could have applied oversampling, which would entail replicating the minority samples until fracture and non-fracture events are well-balanced. In our study, this would require n=120,189 minority cases – i.e. we would need to replicate each minority case almost 16 times – to reach the desired ratio of 3:7. Drawing each minority case that frequently substantially increases the risk of overfitting. Moreover, given our sample size, oversampling would drastically increase the computational runtime and the probability for running out of memory. Consequently, we decided on random undersampling, although randomly drawing from the majority sample sometimes discards informative cases (4, 5). Furthermore, it can distort significance tests and the magnitude of the parameter estimates – i.e. the obtained estimates and p-values ought to be interpreted with caution. In the future, it might be a promising alternative to optimize other metrics than the global classification accuracy, because non-decomposable functions such as the F-Score or precision are more natural choices for imbalanced data that do not require random over- or undersampling (6). Unfortunately, current implementations are limited to using these functions as evaluation criteria, which does not enable parameter optimization.

g) Have you tried to drop the sampling strategy and tried to optimize the F1-Score instead? The F1-Score is independent of balancing. 

The F1-Score is a non-decomposable loss functions – i.e. the loss on a set of points cannot be expressed as the sum of losses on individual data points. This makes it a lot more difficult to implement the F1-Score as a loss function (6). To date, there is no r package that accomplishes this for the algorithms used in our super learner. Instead, the F1-Score is mainly used as an evaluation metric. XGboost has an option to implement early stopping based on the F1-Score, which could potentially help to find a model that performs better on the F1-Score. However, when we implemented this model for our data, it did not converge or pick up any parameter trends towards a global minimum. 

h) Please do add the numbers of false positive and false negative in the validation set.

We added graphs in the supplement that show the development of the proportion and number of false positives and negatives in the validation dataset as a function of the chosen cutoffs for the superlearner model and XGBoost.

We refer to these graphs in the revised manuscript:

Results are shown in Table 2. Furthermore, we provide information on the number of false negatives and false positives as well as the corresponding rates for the superlearner and our benchmark model XGBoost in the supplemental figures 1 and 2.

i) Think of including interactions (with age or gender) in your regression models.

Thank you for this suggestion. We added a logistic regression with these interactions as another candidate algorithm.

j) 65: Germany does not have one of the highest hip-fracture rates – its rather midfield, especially in the 10y probability of major fractures. Better refer to the high rates in Northern Europe (Denmark, Sweden, Norway).

We changed the sentence to:Some of the highest hip fracture rates are seen in Europe and particularly in northern Europe – i.e. Denmark, Sweden and Norway.

k) 66: In the original paper it is 610,000. 

Thank you for pointing that out. We changed the incident number.

l)

68: Punctuation error before reference number

82: Cox proportional hazards regression 

Thank you for pointing that out. We corrected these mistakes.

m)

86-87: What do you mean with ‘advanced’ ML methods – please name those 

We refer to methods which have proven useful over time in kaggle competitions etc. (e.g. neural networks, ensembling strategies or gradient boosting). 

Only recently, advanced machine learning methods – e.g., neural networks, ensembling strategies or gradient boosting – have begun to be used for clinical prediction models (7, 8).

n)

87: ‘may have the potential’ – How large is the expected effect on the AUC? Please refer to other papers on disease predictions where those new methods outperformed classical approaches. 

This is a rather difficult question to answer, because most studies that use machine learning do not compare their chosen algorithm to traditional approaches. In addition, the expected increase in the AUC depends on the number of interactions and non-linear relationships not captured by traditional approaches. 

Nonetheless, we reviewed recent comparisons and referenced these papers in the introduction:

To date, it is difficult to estimate the potential of these techniques, because they are rarely systematically compared to traditional approaches. Furthermore, the utility largely depends on where and how they are employed. Miotto, Li (9) recently showed that unsupervised feature learning based on neural networks can significantly boost the consecutive disease classification from an AUC of 0.632 (worst alternative) to 0.773. However, other studies report rather small improvements in the AUC of around 3% when comparing modern machine learning techniques to traditional approaches (10, 11). Thus, we believe that it is important to examine whether applying more complex algorithms as opposed to traditional regression techniques offers incremental value in various contexts to learn when this additional effort amounts to meaningful improvements.

o)

97: The present study is 

Thank you for pointing this out.

p)

109: Is this the LKK (Landwirtschaftliche Krankenkasse)? I havn’t found ‘German agricultural sickness fund’. 

The study was based on data provided by the German social insurance for agriculture, forestry and horticulture (Sozialversicherung für Landwirtschaft, Forsten und Gartenbau, SVLFG). The LKK was integrated in the SVLFG in spring of 2013.

We added this information under the sample section:

All data were provided by the German agricultural sickness fund Sozialversicherung für Landwirtschaft, Forsten und Gartenbau (SVLFG).

q)

143: How many folds have you used? Define k. 

K=10. 

We applied 10-fold-cross-validation, which divided the dataset into k=10 mutually exclusive and exhaustive sets of almost equal size.

167: SVM already abbreviated

Thank you. We use the abbreviation in the revised version. 

170: Which kernel have you used?

We used a Gaussian radial basis kernel. This information is provided in the supplement.

171: RUSBoost is an abbreviation, please add also the full name. 

We added the full name:

We considered the following candidate learning algorithms: Logistic regression using forward variable selection (12), random forests (13), support vector machines (SVM) (14), RUS (random undersampling) - Boost with SVM as learner (15, 16).

r)

196: Why do you use 3:7 sampling? I there any recommendation giving in the literature? 

The exact ratio is arbitrary. To date, to the best of our knowledge, there are no recommendations or simulations on the best possible ratio in imbalanced datasets with different sample sizes. A ratio of 50:50 is ideal for algorithms, because the error on minority samples (i.e. patients with a hip fracture) is as important for minimizing the loss function as the error on the majority sample. However, when random undersampling is applied, this leads to discarding a lot of potentially useful majority samples. If we would have used a 50:50 ratio, our sample size would have decreased to n=12,274 as opposed to n=20,456. The ratio of 3:7 offers both near balance of majority to minority cases and an acceptable loss of information – given the large number of majority cases. 

Other comments:

219: I just see 20,456 individuals used for training and 20% of 288,000 for testing. To the analysis did not include 288,086 individuals. 

We agree that the wording was misleading. We changed the sentence:

Of the 288,086 individuals that were originally included in the sample, 20,456 were used during training and 57,618 in the validation dataset. Mean age of the original sample was 75.7 years (SD: 6.2), 48.8% were female.

221: Please write the number in numeric form: 2.4%

223: 0.6% of all samples 

We corrected both aspects.

234: Please add columns with the number of individuals with and without 4yr-hip-fracture to allow univariable comparisons (and probably important for inclusion in meta-analysis).

Thank you for pointing that out. We added this information to table 1. In addition, we included the number of patients without a hip fracture within the 4 year follow-up who were not lost to follow-up.

249: The MSE for SVM can be calculated in R using caret, see https://www.quora.com/How-can-I-calculate-the-mean-square-error-MSE-for-SVM-in-R

Thank you for this suggestion, the caret package offers several valuable features. However, caret does not evaluate the RMSE for classification models, which completely makes sense. We made a mistake by writing that we calculated the MSE, because we calculated the brier score. It is mathematically really similar, but determines the difference between the estimated probabilities and the actual outcome (1 for events and 0 for non-events). The brier score should not necessarily be reported for support vector machine output, because SVMs do not output probabilities (17). Nonetheless, we used an additional sigmoid function to map the traditional output of the SVM to probabilities (although these are often not well calibrated (18)). We report the brier score based on the transformed output of the SVM. Moreover, we changed the caption und the legend of table 2, so that it is clear that the output of the SVM was transformed before calculating the brier score.

257: Suggested section title: Predictors 

We agree that this keyword better captures the content of the section.

267: Table header suggestion: Coefficients of multivariable logistic regression to predict hip fracture

267: Please add the p-values.

267: Please add the unit of ‘Age’ to make clear that this is the beta-coefficient for 1 year difference.

Thank you for these suggestions. We changed the table accordingly. 

282: Full stop after algorithms and add citation 

We changed the sentence:

Notably, in several datasets the super learner increased performance when compared to its candidate algorithms (13). Nevertheless, we found multiple studies, who correspondingly found no major differences between the super learner and more traditional methods or even somewhat worse performance (34, 35). 

293: I disagree, this is not computationally expensive – it can be directly extracted from the model

At the time of analysis, it was still relatively difficult to obtain the variable importance plot for certain algorithms, because this functionality had not been implemented yet in all r packages. However, we agree that the caret and XGBoost library make this process easy. Moreover, it is no longer computationally demanding for most algorithms. Nonetheless, it can be time-consuming, because calculating the variable importance requires some-sort of cross validation to obtain an out of the bag sample and iteratively shuffling each predictor variable in the validation dataset during prediction. This can be computationally expensive if the libraries are not written in C++ or Fortran. In addition, the superlearner is constructed after the cross-validation results are obtained, which makes it difficult to assess variable importance during training. 

Nonetheless, we agree that this is no longer a valid argument for not providing any variable importance plot. Thus, we added a variable importance from the XGBoost algorithm in the supplement and deleted the paragraph on the computational expense of calculating variable importance. 

As it is recommended to decide upon input variables, candidate algorithms, and their specification before running the analysis (11), we considered risk factors commonly included in fracture risk assessment tools that were also available in claims data. To help to achieve a better understanding of important risk factors, we reported the output of a logistic regression analysis and report a variable importance plot of XGBoost in the supplement.

302: Please compare your results with described performances in the literature, … AUC in model with bone density information is 0.82 …

We added a section that highlights the utility of having clinical and laboratory information for hip fracture predictions:

Regarding our data source, we found that predictions based on german claims data are considerably worse than predictive models that take advantage of clinical information such as bone mineral densities and biochemical glucose measurements. In Denmark, a recent study reported an AUC of up to 0.92 for hip fracture predictions, which highlights the benefit of using a National Patient Registry that contains clinical and laboratory information for risk prediction (41). 

306: Please add citation 

We rewrote the section and added citations:

The fact that drug-related risk factors had limited additional explanatory power in this study may partly explain moderate performance of the super learner. Our choice to employ a super learner approach in this study was driven by the effectiveness of the technique in other studies (7, 17), the low base rate of hip fractures in our data and the relatively large number of potential predictors we initially considered. Yet, it is still not fully understood whether and when complex ensemble machine learning techniques offer real value over traditional methods in clinical prediction tasks (18). We agree with previous findings on this question that a strong non-linear relationship between the predictors and the response appears to be an important prerequisite (19) and that the signal-to-noise ratio ought to be high (20). 

313: If your intention was to find complex interaction, why do you just present results from the logistic regression without any interaction?

We agree that the wording was misleading. We wanted to take advantage of complex interactions if there were any. We changed the sentence accordingly:

This allowed us to apply powerful machine learning methods for fracture prediction, which have the potential to take advantage of complex interactions and unknown non-linear effects.

However, the fact that we found no difference between techniques that can capture interactions and those that only account for main effects indicates that there are no relevant and robust interactions to report. Nonetheless, we added evaluation criteria for the models with interactions. These show no improvement over the simpler models. Therefore, we report the simplest model.

319: I don’t see any memory problems arising from a data frame with 20000 individuals and probably less than 100 features. 

The sentence may have been unclear. We would have had memory issues if we had worked with the original training data, which contained n=230.468 individuals. The corresponding paragraph was changed to make that more clear.

359: software 

Thank you. We corrected this mistake.

Comments of Reviewer 2:

a) The sentence started from the line 80 “ Moreover, claims data in Germany ……within the observational period” was not clear, please clarify 

Thank you for pointing this out. We elaborated on the statement:

Moreover, claims data in Germany include information, which can be difficult to obtain when questionnaires or interviews are used, because – as opposed to questionnaires and interviews – the data is not prone to recall (e.g. patients who do not remember their prescription dates) or recruitment biases (i.e. the full cohort of insuree’s and not just those who choose to participate are assessed).

b) In the paragraph of “sample”, the baseline for the study was not well defined. For an example, in line 108, the baseline is April, 1 2010 and pre baseline is 24 months. Were patients allowed to enter the cohort after April, 1, 2010? 

We rephrased the paragraph to make it more clear that patients who switched to the social insurance during the pre-period or after April, 1, 2010 were excluded:

Notably, the dataset is limited to people working in agriculture and their families, because the data provider is the German agricultural sickness fund – in german: Sozialversichung für Landwirtschaft, Forsten und Gartenbau (SVLFG). We only included individuals, who were insured by the SVLFG on April 1, 2010 (baseline) with continuous insurance coverage for the 24 months pre-period – i.e. we excluded patients who switched to the SVLFG during the pre-period or after April, 1, 2010.

c) In the paragraph of “ Outcome variable”, “hip fracture occurring within 4 years of baseline” was the outcomes of interest? Please clarify baseline and prebaseline. 

In the revised manuscript, we mention the exact time period:

The outcome variable was the first hip fracture occurring within 4 years after the index date – i.e. between April 1 2010 and 31 March 2014.

d) It was not clear whether the patients were allowed to have multiple fractures or only the first one. We assessed the delay between the index date and the first fracture that occurred. Thus, we did not differentiate between a patient that received multiple fracture in the observational period or one fracture. As a predictor, we included the information, whether a patient had at least 1 fracture in the pre-period. Consequently, it was allowed to have multiple fractures within the preperiod, but it was not explicitly accounted for in our model.

We changed two relevant sentences to make this more clear:

We further assessed whether at least one prior fracture within 2 years preceding baseline was recorded (yes/no).

The outcome variable was the first hip fracture within 4 years.

e) Please explain why some variables were evaluated using 2 years pre baseline, but some factors were evaluated in the seven month before baseline. 

We differentiated between predictors related to the individual fracture history and predictors related to current medications used. 

Regarding the fracture history, an association between prior fractures and fracture risk is well established, but in previous validation studies – e.g. during the validation of QFracture (21) – it was not explicitly assessed whether the predictive validity of prior fractures depends on the recency of the prior fracture. Thus, we used the entire available preperiod to detect previous fractures. 

Regarding risk factors related to medication use, it was found that the recency of the prescribed medication matters when predicting fracture risk. For instance, for oral corticosteroids, it was observed that fracture risk increases rapidly after the commencement of oral corticosteroid therapy, but reverses sharply toward baseline levels after discontinuation of oral corticosteroids (22). Consequently, a two year preperiod for assessing medication use would be too long, because insurees may have already stopped taking these medications. 

f) This paper used prescribed medications as a surrogate for diseases associated with fracture risk, however, many medications have multiple indications. Has these surrogates been validated yet? 

Thank you for pointing that out. Maybe it was unclear in our previous draft that not all of the assessed prescriptions were surrogates for diseases. We agree that some of the medications we included have multiple indications. However, we also wanted to include medications that have multiple indications but were consistently associated with fracture risk in previous studies such as psychotropic drugs (23). In the revised version of the manuscript we differentiate between reliable surrogate variables (e.g. antiparkinson agents) and simple medication based risk factors.

We recognise that the ascertainment of (co-)morbidities proves a challenge in (administrative) ambulatory claims data. The lack of reliability of diagnosis coding in the ambulatory setting has been repeatedly shown (24, 25). Therefore, we solely considered drug exposure as surrogate variable, but no ambulatory ICD-10 diagnoses to identify and define conditions that have been associated with increased fracture risk such as Parkinson's disease (26). Prescribed medications whose ATC Classification starts with N04 are almost exclusively used to manage Parkinson's disease and are therefore a reliable indication for that particular disorder. This prescription based measure was chosen, because using the most reliable source of information (i.e. hospital diagnoses) would have introduced considerable bias, because in this case only individuals with a hospital stay would have been recorded. In addition, we included medications that have multiple indications but were consistently associated with fracture risk in previous studies such as psychotropic drugs (23). 

g) Suggest to also use logistic regression backward as it is has been reported more conservative 

Thank you for this suggestions. We reported the results on the logistic regression with backward selection as well. Moreover, we revised the superlearner model. In the revised version of the manuscript, the superlearner considers both the logistic regression with backward and forward selection as candidate algortihms. 

We also followed the suggestions made by Reviewer 1 to include an alternative specification of the regression models with interactions (with age or gender). Our revised superlearner model chose the logistic regression with interactions and backward selection as the most suitable candidate among the four alternative specifications. 

h) The analysis section is clear and well written 

Thank you.

i) Suggest the authors to add the time period for the medications in table 1 

We added the time period to table 1. 

j) Please also provide the characteristics for patients risk during the follow up time. 

We are uncertain whether we understand this comment correctly. It is unconventional to provide sample characteristics for the follow up period as well. Usually, the baseline characteristics are provided to describe the sample for which outcomes are assessed. However, we added the number of patients that switched to a different statutory health insurance and those that were lost to follow-up due to death, because these characteristics might influence the performance of the prediction model. 

Literature:

1. Babyak MA. What you see may not be what you get: a brief, nontechnical introduction to overfitting in regression-type models. Psychosomatic medicine. 2004;66(3):411-21.

2. Mangal A, Kumar N, editors. Using big data to enhance the bosch production line performance: A kaggle challenge. 2016 IEEE International Conference on Big Data (Big Data); 2016: IEEE.

3. Georganos S, Grippa T, Vanhuysse S, Lennert M, Shimoni M, Wolff E. Very high resolution object-based land use–land cover urban classification using extreme gradient boosting. IEEE geoscience and remote sensing letters. 2018;15(4):607-11.

4. Yap BW, Rani KA, Rahman HAA, Fong S, Khairudin Z, Abdullah NN, editors. An application of oversampling, undersampling, bagging and boosting in handling imbalanced datasets. Proceedings of the first international conference on advanced data and information engineering (DaEng-2013); 2014: Springer.

5. Batuwita R, Palade V, editors. Efficient resampling methods for training support vector machines with imbalanced datasets. The 2010 International Joint Conference on Neural Networks (IJCNN); 2010: IEEE.

6. Eban EE, Schain M, Mackey A, Gordon A, Saurous RA, Elidan G. Scalable learning of non-decomposable objectives. arXiv preprint arXiv:160804802. 2016.

7. Pirracchio R, Petersen ML, Carone M, Rigon MR, Chevret S, van der Laan MJ. Mortality prediction in intensive care units with the Super ICU Learner Algorithm (SICULA): a population-based study. The Lancet Respiratory Medicine. 2015;3(1):42-52.

8. Ma F, Chitta R, Zhou J, You Q, Sun T, Gao J, editors. Dipole: Diagnosis prediction in healthcare via attention-based bidirectional recurrent neural networks. Proceedings of the 23rd ACM SIGKDD international conference on knowledge discovery and data mining; 2017.

9. Miotto R, Li L, Kidd BA, Dudley JT. Deep patient: an unsupervised representation to predict the future of patients from the electronic health records. Scientific reports. 2016;6(1):1-10.

10. Acion L, Kelmansky D, van der Laan M, Sahker E, Jones D, Arndt S. Use of a machine learning framework to predict substance use disorder treatment success. PloS one. 2017;12(4):e0175383.

11. Weng SF, Reps J, Kai J, Garibaldi JM, Qureshi N. Can machine-learning improve cardiovascular risk prediction using routine clinical data? PloS one. 2017;12(4).

12. Hosmer D, Lemeshow S. Applied Logistic Regression 2nd edn Wiley-Interscience Publication. John Wiley, Hoboken, New Jersey; 2000.

13. Breiman L. Random forests. Machine learning. 2001;45(1):5-32.

14. Vapnik VN. Statistical learning theory: John Wiley & Sons, New York. A Wiley-Interscience Publication; 1998.

15. Seiffert C, Khoshgoftaar TM, Van Hulse J, Napolitano A. RUSBoost: A hybrid approach to alleviating class imbalance. IEEE Transactions on Systems, Man, and Cybernetics-Part A: Systems and Humans. 2010;40(1):185-97.

16. Seiffert C, Khoshgoftaar TM, Van Hulse J, Napolitano A, editors. RUSBoost: Improving classification performance when training data is skewed. Pattern Recognition, 2008 ICPR 2008 19th International Conference on; 2008: IEEE.

17. Polley EC, Van der Laan MJ. Super learner in prediction. 2010.

18. Jie M, Collins GS, Steyerberg EW, Verbakel JY, van Calster B. A systematic review shows no performance benefit of machine learning over logistic regression for clinical prediction models. Journal of clinical epidemiology. 2019.

19. Dorie V, Hill J, Shalit U, Scott M, Cervone D. Automated versus do-it-yourself methods for causal inference: Lessons learned from a data analysis competition. Statistical Science. 2019;34(1):43-68.

20. Ennis M, Hinton G, Naylor D, Revow M, Tibshirani R. A comparison of statistical learning methods on the GUSTO database. Statistics in medicine. 1998;17(21):2501-8.

21. Hippisley-Cox J, Coupland C. Derivation and validation of updated QFracture algorithm to predict risk of osteoporotic fracture in primary care in the United Kingdom: prospective open cohort study. 2012.

22. Van Staa T, Leufkens H, Abenhaim L, Zhang B, Cooper C. Use of oral corticosteroids and risk of fractures. Journal of bone and mineral research. 2000;15(6):993-1000.

23. Study HF, Grisso JA, Kelsey JL, O'Brien LA, Miles CG, Sidney S, et al. Risk factors for hip fracture in men. American Journal of Epidemiology. 1997;145(9):786-93.

24. Wilchesky M, Tamblyn RM, Huang A. Validation of diagnostic codes within medical services claims. Journal of clinical epidemiology. 2004;57(2):131-41.

25. Erler A, Beyer M, Muth C, Gerlach F, Brennecke R. [Garbage in-garbage out? Validity of coded diagnoses from GP claims records]. Gesundheitswesen (Bundesverband der Arzte des Offentlichen Gesundheitsdienstes (Germany)). 2009;71(12):823-31.

26. Grisso JA, Kelsey JL, Strom BL, Ghiu GY, Maislin G, O'Brien LA, et al. Risk factors for falls as a cause of hip fracture in women. New England journal of medicine. 1991;324(19):1326-31.

---

## [Decision Letter · Decision Letter 1]

27 Apr 2020

Osteoporotic hip fracture prediction from risk factors available in administrative claims data – a machine learning approach

PONE-D-19-26721R1

Dear Dr. Engels,

We are pleased to inform you that your manuscript has been judged scientifically suitable for publication and will be formally accepted for publication once it complies with all outstanding technical requirements.

With kind regards,

Lars Kaderali

Academic Editor

PLOS ONE

Additional Editor Comments (optional):

Reviewers' comments:

Reviewer's Responses to Questions

**Comments to the Author**

1. If the authors have adequately addressed your comments raised in a previous round of review and you feel that this manuscript is now acceptable for publication, you may indicate that here to bypass the “Comments to the Author” section, enter your conflict of interest statement in the “Confidential to Editor” section, and submit your "Accept" recommendation.

Reviewer #1: All comments have been addressed

2. Is the manuscript technically sound, and do the data support the conclusions?

Reviewer #1: Yes

3. Has the statistical analysis been performed appropriately and rigorously? 

Reviewer #1: Yes

4. Have the authors made all data underlying the findings in their manuscript fully available?

Reviewer #1: Yes

5. Is the manuscript presented in an intelligible fashion and written in standard English?

Reviewer #1: Yes

6. Review Comments to the Author

Reviewer #1: Thank you for addressing all my comments, questions and remarks and making appropriate changes to the manuscript.

Minor typos / corrections:

81: Please write 'area under the receiver operating characteristic curve (AUC)' to introduce the abbreviation instead of line 201

144: 10-fold cross-validation

145: remove space before comma

164: gold standard

247: Lost to follow-up,

death within four years,

Death within the first year,

Death within two years,

Death within three years,

Death within four years,

51,476

250: With regard to the Brier score,

252: Brier score

256ff: Stick to one notation of either '95%CI', '95% CI', or '95%-CI', the same accounts to 'super learner'/'superlearner'

264: Brier score

267: dot before 'Thus'

268/270: Chi-Square instead of x²

Typos/grammar in the supplement:

p1/RF: An ensemble method

p1/RF: remove additional white space before 'on'

p1/LR and RF: add dot at the end of the sentence

p2: better: 'otherwise the model is likely to be adapted too much to the training data and is therefore not generalized to the validation data set'

p2: 10-fold cross-validation

p2: Table S2

p2: subsample=0.8

p3: prediction

p4: prediction

p5: distinguish

7. PLOS authors have the option to publish the peer review history of their article (what does this mean?). If published, this will include your full peer review and any attached files.

Reviewer #1: Yes: Marcus Vollmer

---

## [Editor Report · Acceptance letter]

8 May 2020

PONE-D-19-26721R1 

Osteoporotic hip fracture prediction from risk factors available in administrative claims data – a machine learning approach 

Dear Dr. Engels:

I am pleased to inform you that your manuscript has been deemed suitable for publication in PLOS ONE. Congratulations! Your manuscript is now with our production department. 

With kind regards,

on behalf of

Prof. Dr. Lars Kaderali 

Academic Editor

PLOS ONE